# BEYOND URLS: METADATA DIVERSITY AND POSITION FOR EFFICIENT LLM PRETRAINING

**Dongyang Fan**⋆♡**, Diba Hashemi**⋆♡**, Sai Praneeth Karimireddy**◇**, Martin Jaggi**♡
♡ EPFL, ◇ University of Southern California
Correspondence: {dongyang.fan,diba.hashemi}@epfl.ch
⋆ denotes equal contribution

## ABSTRACT

Incorporating metadata in Large Language Models (LLMs) pretraining has recently emerged as a promising approach to accelerate training. However prior work highlighted only one useful signal—URLs, leaving open the question of whether other forms of metadata could yield greater benefits. In this study, we investigate a wider range of metadata types and find other types of metadata, such as fine-grained indicators of document quality that can also accelerate pretraining when prepended. We identify a common feature among effective metadata: they encode information at a finer granularity. We further introduce metadata appending as a means of improving training efficiency, where predicting an appropriate metadata as auxiliary task can help speed up pretraining. In addition, learnable meta-tokens trained with masked loss can recover part of the speedup by inducing quality-aware latent structure. Using probing, we analyze latent representations to understand how metadata shapes learning. Together, these results yield practical guidelines for integrating metadata to improve both the efficiency and effectiveness of LLM pretraining.

## 1 INTRODUCTION

Large language models (LLMs) are typically pretrained on web-scale corpora sourced from Common Crawl–style snapshots and related aggregates, then aggressively filtered and deduplicated to improve quality and efficiency. Landmark datasets such as C4 (Raffel et al., 2020) and RefinedWeb (Penedo et al., 2023) exemplify this trend: both begin from massive web crawls and rely on heuristic and statistical filtering to remove boilerplate, low-quality, and near-duplicate content before tokenization; more recent open corpora like Dolma (Soldaini et al., 2024a) extend this paradigm to trillions of tokens spanning diverse sources. In parallel, a growing literature studies data filtering/selection, e.g., perplexity- or importance-based selection, to allocate pretraining compute toward the most useful subsets, for which we refer the readers to a survey (Albalak et al., 2024). Together, these efforts frame pretraining efficiency largely as a problem of "which web data to keep and how much of it to use."

A complementary axis for improving pretraining efficiency is to enrich the input with document-level metadata, such as information about a text's source, domain, time, or other attributes, so the model can condition its representation learning accordingly. Recent work formalizes this idea as "metadata conditioning", showing that prepending simple, readily available indicators (e.g., source URLs or domain tags) during pretraining can accelerate learning and improve downstream controllability, with a "cool-down" phase to remove the dependency at inference (MeCo, Gao et al. (2025)). This direction builds on the broader intuition that structural signals beyond raw tokens (such as links, domains, registers) can shape what is learned during pretraining.

However, the landscape of effective metadata types and positions remains underexplored. To date, empirical evidence most consistently supports prepending the URL (or source identifier) before the document as a practical and robust way to accelerate LLM pretraining; systematic comparisons further report that other readily available metadata (e.g., coarse topics or quality indicators) have not yet shown comparable acceleration under similar budgets and setups (Fan et al., 2025). Whether richer metadata schemas (beyond URLs) or alternative integration strategies (e.g., suffixing, special-token segment headers, side-channels) yield additional efficiency gains is still largely an open question.

Finally, despite encouraging empirical gains, we have limited mechanistic understanding of how metadata shapes latent representations during pretraining, e.g., whether conditioning induces domain-specific subspaces, affects token- and document-level mutual information, or alters cross-domain interference and transfer. Preliminary analyses from metadata-conditioning hint that such signals can reorganize representation geometry and influence downstream behavior, but a principled account connecting metadata types, injection positions, and emergent representation structure remains to be developed.

We summarize our contributions as follows:

- We identify additional useful metadata signals in accelerating LLM pretraining when prepended, beyond URL. We show that the fine-granularity of the metadata is the key in bringing the acceleration effect (Section 4.1).

- We introduce and evaluate metadata appending as a method for accelerating pretraining, and highlight metadata types that are particularly suitable as auxiliary signals in this setup (Section 4.2).

- We show that learnable meta tokens can partially recover the speedup with metadata, where attention patterns to these tokens encode quality-aware information. (Section 4.4).

- We conduct layer-wise probing of latent representations for topic, quality, and authorship, providing mechanistic insight into how these factors are better encoded in the latent space (Section 4.5).

## 2 RELATED WORK

**Pretraining efficiency of LLMs.** Pretraining efficiency is enhanced through three axes: 1) Better pretraining data. RefinedWeb (Penedo et al., 2023) argues that carefully filtered web-only data can outperform mixed curated corpora, while FineWeb (Penedo et al., 2024) scales this idea to ~15T tokens with stronger decanting and deduplication at crawl-snapshot scale. Both report improved downstream performance per token, i.e., better data efficiency. In parallel, Dolma (Soldaini et al., 2024b) and RedPajama (Weber et al., 2024) foreground transparent releases with quality signals, dedup IDs, and fine-grained metadata to enable learned filtering and weighting strategies—aiming to squeeze more performance out of each pretraining FLOP. 2) More efficient architectural design. Sparse Mixture-of-Experts architectures (e.g., Switch Transformer, Fedus et al. (2022)) activate only a small subset of parameters per token, yielding large effective capacity at near-constant compute and demonstrating multi-× pre-training speedups at scale. Multi-Query Attention (MQA, Shazeer (2019)) and Grouped-Query Attention (GQA, Ainslie et al. (2023)) share or group KV heads to reduce KV cache and bandwidth costs; although often discussed for inference, these changes also lower training-time memory traffic and can improve throughput for long contexts. 3) Additional signals from metadata. Recently, Gao et al. (2025); Fan et al. (2025) have both demonstrated an acceleration effect from URL prepending, saving up to 30-40% tokens in pretraining time. This acceleration is considerable, even on top of data filtering. In this work, we aim to further advance this axis.

**Metadata in LLM pretraining.** A growing body of work shows that exposing explicit metadata, e.g., URL/domain, document IDs, and timestamps, can improve efficiency, steerability, and attribution. On the source/domain axis, CTRL conditions on control codes derived from data structure (domain, dates, etc.) to steer generation (Keskar et al., 2019); similarly, Fan et al. (2025) find that prepending topic and format tokens yield stronger controllability than raw URL domains. For provenance and attribution, Source-Aware Training injects document IDs in a light post-pretraining stage to enable intrinsic citation of pretraining sources (Khalifa et al., 2024). Along the temporal dimension, Zhao et al. (2024) align pretrained LMs to a target year using timestamp metadata, and Faro et al. (2025) pretrain experts on time-sliced corpora with routing by query time, boosting time awareness while preserving downstream performance. Finally, a line of metadata-as-context work shows theoretically and empirically that prepending metadata tokens (excluded from the loss) can substantially accelerate pretraining (Allen-Zhu & Li, 2024; Gao et al., 2025; Zhu et al., 2025; Fan et al., 2025). However, the benefits of metadata conditioning is not uniform: effectiveness depends on prompt length and setting (Higuchi et al., 2025; Fan et al., 2025). We extend this line by examining efficiency gains

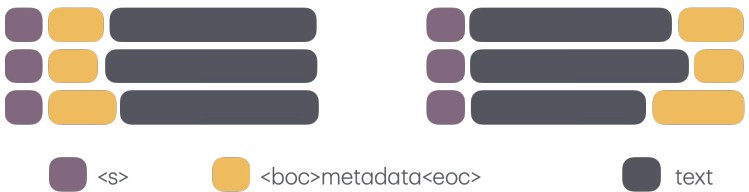

Figure 1: Diagram of our tokenization. Each document begins with a default beginning-of-sequence (``) token. For each sequence, metadata is wrapped between beginning-of-context (`<boc>`) and end-of-context (`<eoc>`). Depending on the metadata position, the metadata is prepended (illustrated on the left) or appended (illustrated on the right) to the document. If a long document split into multiple sequences, metadata is attached to each one. A 10% dropout of metadata is always performed.

from metadata appending and by introducing learnable metatokens as a flexible alternative to fixed metadata strings.

## 3 EXPERIMENTS

We follow the experimental setup of Fan et al. (2025), and extend their investigation to more metadata types. The model is an adopted 1.5B Llama model with 16 layers in total. We use AdamW optimizer with regularization strength 0.1 (Loshchilov & Hutter, 2019). For each batch, the model processes 2.06 million tokens, corresponding to a sequence length of 4,096 and a batch size of 504. We use cosine scheduler with a maximum learning rate of 3e-4, and in the first 5% steps we perform the learning rate warm-up. To train the models, we use the Megatron-LM framework (Narayanan et al., 2021).

We use FineWeb-Edu (Lozhkov et al., 2024) for comparability with Fan et al. (2025). In the metadata-prepending setup, we mask the metadata tokens from the loss both for reporting and during backpropagation. In the metadata-appending setup, we mask the metadata only for loss reporting; we retain its loss for backpropagation so the model learns to predict the metadata. The tokenization approach for metadata prepending and appending is depicted in Figure 1.

**Evaluation Benchmarks**. As standard practice, we evaluate models on general knowledge understanding using LM-Eval-Harness developed by Gao et al. (2024). The benchmarks used are Arc-Easy (Clark et al., 2018), Arc-Challenge (Clark et al., 2018), CommonSense QA (CSQA, Talmor et al., 2018), MMLU (Hendrycks et al., 2020), PIQA (Bisk et al., 2020), Social IQA (SIQA, Sap et al., 2019), HellaSwag (HS, Zellers et al., 2019), Lambada (LBD, Paperno et al., 2016) and Winogrande (WG, Sakaguchi et al., 2021).

**Metadata Types.** We examine the following types of metadata. Building on Fan et al. (2025), we extend the quality score and domain information to a finer level of granularity. In addition, for each metadata type, we explore two positional variants: prepending and appending.

- *Full URL (URL)*: Provided by the FineWeb-Edu dataset as `url`, from which the document was crawled.

- *Coarse-Grained Quality Score (QS-coarse)*: Provided by the FineWeb-Edu dataset as `int_score`. These scores are generated by a linear regressor trained on 410,000 web samples, each annotated by Llama-3-70B-Instruct on a scale from 0 (not educational) to 5 (highly educational), reflecting the sample's educational value. The scores are then rounded to the closest integers, and only scores greater than or equal to 3 are kept. There are 3 categories of coarse-grained quality levels.

- *Fine-Grained Quality Score (QS-fine)*: Similar to QS-coarse, we take the raw score (provided as `score`) from the regressor instead of the rounded one. The score is presented as $\lfloor \texttt{score} * 10 \rfloor$. Thus, it is at least 10 times finer than QS-coarse.

- *Coarse-Grained Domain Information (DI-coarse)*: Following Fan et al. (2025), we annotate each document with WebOrganizer (Wettig et al., 2025), where topic and format domains

are returned by pretrained classifiers. There are 24 categories per taxonomy. In total, we have 576 different Domain Information types.

- *Fine-Grained Domain Information (DI-fine)*: Each document is labeled with topic and format domains generated by Llama3.1-8B-Instruct model (Grattafiori et al., 2024). The generation prompt can be seen in Appendix C. The generation is open-ended, and there is are unlimited number of categories.

- *Meta Tokens*: Five empty different meta tokens newly added to the vocabulary. The usage is detailed in Section 4.4.

**Probing Experiments.** In addition to downstream evaluation benchmarks, we employ probing to analyze the information encoded in layerwise representations. While benchmarks reflect the model's overall knowledge and reasoning ability, they provide limited insight into how metadata affects the internal learning dynamics of different layers. Probing allows us to examine these differences more directly and shed light on the representational role of metadata in accelerating pretraining.

Specifically, we train probing classifiers on the following tasks:

- *Document quality prediction.* We sample 15,000 documents from FineWeb-Edu, annotated with QS-coarse and balanced across three quality levels (5,000 per score). Using a 90/10 train–test split, the objective is to predict the quality score from the document representation.

- *Document topic prediction.* We sample 20,000 documents from FineWeb-Edu, annotated with DI-coarse topics and balanced across 20 categories (1,000 per topic). Using the same 90/10 train–test split, the goal is to predict the topic label from the document representation.

- *Authorship prediction.* We use the dataset from Stamatatos (2017), consisting of 2,157 Guardian articles written by 13 authors. We adopt a 70/30 train–test split. This task is relatively straightforward: later layers achieve 100% test accuracy, while earlier layers still retain some distinguishable signal.

For these tasks, we train layer-wise three-layer MLP classifiers using representations from each layer. The probe takes the last hidden state of each document as input (truncated at the 100th token); for shorter documents, the final token representation is used instead.

## 4 How can metadata enhance pretraining?

In this long section, we present results from a thorough investigation of metadata types and positions. Throughout, we explain some unique phenomena by probing through the latent representations.

### 4.1 Prepending: Training Speed-up with fine-grained metadata conditioning

On top of the metadata types examined by Fan et al. (2025), we extend the investigation to two other fine-grained metadata types. The left panel of Figure 2 shows how downstream performances progress with prepending different metadata. The final average performance across downstream tasks is reported in Table 1. Among the five context types tested, both URL and fine-grained quality scores yield comparable acceleration, reaching the performance of a 100B-token baseline after only 60B tokens. Fine-grained domain metadata also provides a boost, surpassing the 100B-token baseline with 20B fewer tokens. It seems that quality score and domain information can be helpful, but fine granularity is the key.

> **Observation 1.** Only fine-grained metadata conditioning has a positive effect in speeding up pretraining; conditioning on coarse-grained meta-information yields no noticeable change.

**No additive effect from metadata.** We further examine whether the model benefits from prepending two types of helpful metadata. Specifically, we provide both the URL and the fine-grained quality score as metadata and compare the results in Figure 3. The model shows faster learning during the early stages of training, indicating that the combined metadata helps it acquire information more quickly. However, as training progresses and more tokens are seen, this advantage diminishes, and

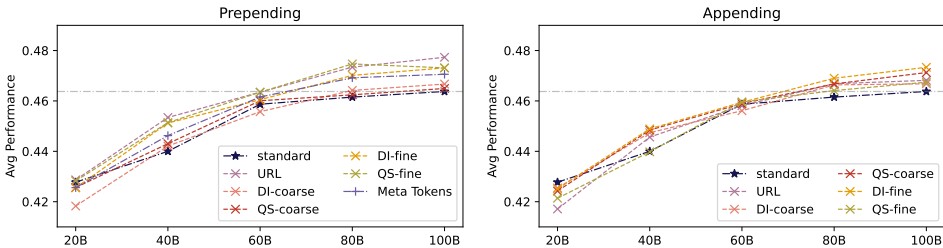

Figure 2: Pretraining acceleration is measured using downstream evaluation performance. The dashed horizontal line marks the downstream performance achieved after 100B tokens of standard pretraining. DI denotes domain information, and QS denotes quality score; the "-fine" and "-coarse" suffixes indicate fine-grained and coarse-grained variants, respectively. **Left:** Prepending fine-grained metadata matches the 100B-token standard-pretraining performance using only 60B tokens. **Right:** Appending fine-grained domain information and a coarse-grained quality score matches the 100B-token baseline using 80B tokens.

Table 1: Evaluation results on the average of 9 downstream tasks. When evaluating prepending models, we add the prefix `<boc><eoc>` to each evaluation text. When evaluating appending models and the standard model, we add no prefix texts. For every configuration that outperformed standard training on average, we report results as the mean over three random seeds to reduce variance. DI denotes domain information, and QS denotes quality score.

|  | Arc-C | Arc-E | CSQA | MMLU | PIQA | SIQA | HS | LBD | WG | Avg |
|---|---|---|---|---|---|---|---|---|---|---|
| standard | 33.9 | 68.6 | 45.1 | 31.8 | 71.7 | 41.5 | 41.9 | 30.6 | 54.9 | 46.7 |
| `<boc><eoc>` | 34.0 | 69.2 | 43.5 | 31.9 | 71.9 | 41.5 | 42.5 | 30.8 | 55.2 | 46.7 |
| *prepending* | | | | | | | | | | |
| URL | 34.8 | 71.7 | 46.9 | 32.5 | 72.3 | 41.5 | 42.5 | 32.4 | 55.7 | **47.7** |
| QS (Coarse) | 32.7 | 69.6 | 41.0 | 31.8 | 72.1 | 41.6 | 43.0 | 32.4 | 55.2 | 46.6 |
| QS (Fine) | 35.3 | 70.0 | 45.4 | 32.1 | 72.4 | 41.6 | 42.3 | 32.3 | 54.5 | **47.3** |
| DI (Coarse) | 34.0 | 69.8 | 42.6 | 32.1 | 71.9 | 40.3 | 42.0 | 31.8 | 55.5 | 46.7 |
| DI (Fine) | 34.7 | 70.5 | 46.0 | 32.6 | 72.8 | 40.8 | 42.1 | 32.1 | 54.5 | **47.3** |
| Meta Tokens | 35.1 | 69.8 | 45.0 | 32.1 | 72.1 | 40.1 | 42.0 | 32.1 | 55.3 | 47.1 |
| *appending* | | | | | | | | | | |
| URL | 34.3 | 70.1 | 44.6 | 32.0 | 72.0 | 40.7 | 41.9 | 31.2 | 54.9 | 46.8 |
| QS (Coarse) | 34.4 | 69.0 | 46.0 | 32.1 | 72.6 | 40.6 | 42.3 | 31.1 | 56.2 | 47.1 |
| QS (Fine) | 34.8 | 69.8 | 44.9 | 32.2 | 71.8 | 36.5 | 42.2 | 33.0 | 55.6 | 46.8 |
| DI (Coarse) | 33.8 | 68.7 | 44.6 | 31.7 | 71.9 | 40.5 | 41.9 | 32.3 | 54.9 | 46.7 |
| DI (Fine) | 34.7 | 71.3 | 45.3 | 32.1 | 71.2 | 41.6 | 42.3 | 34.4 | 53.2 | **47.3** |

the final performance is comparable to that of a model trained without metadata. Notably, the model does not appear to leverage either the URL or the fine-grained quality score consistently.

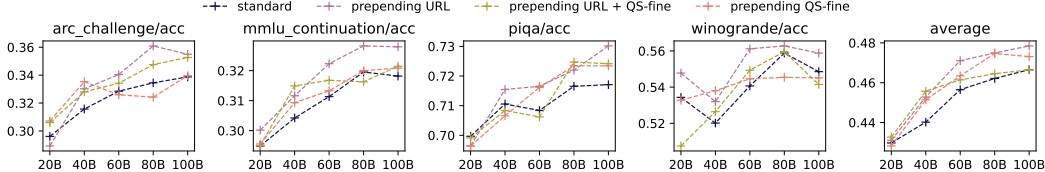

Figure 3: Comparison of downstream performance when prepending URL or fine-grained quality score (QS-Fine) individually versus in combination. Both metadata types are effective on their own, but combining them yields no additive effect on metadata acceleration.

**Fine-grained metadata prepending performs better than coarse-grained.** In Figure 2 and Table 1, we observe that the fine-grained metadata types lead to stronger model performance compared to their coarse-grained counterparts. We hypothesize that more fine-grained metadata distinctions

improve the model's ability to capture and represent salient information in the data. To test this hypothesis, we probe document-topic prediction using representations from two models trained with either fine- or coarse-grained metadata prepended, as described in Section 3. As shown in Figure 4, prepending either fine- or coarse-grained metadata improves performance over the standard model. The fine-grained variant yields a modest but consistent gain, suggesting that fine-grained metadata prepending helps the model encode topic information more effectively in its latent representations. . To verify that our probing procedure is sound, we provide further evidence in Appendix B.4.

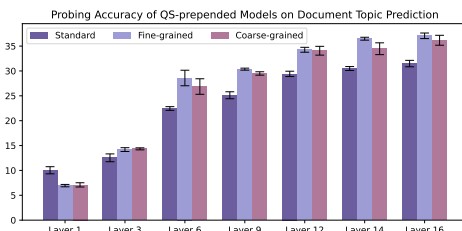 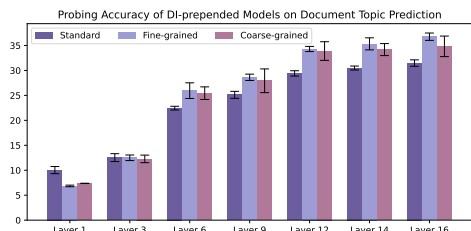

Figure 4: Probing results on document topic prediction for models prepended with quality scores (QS) and domain information (DI), with fine- and coarse-grained variants. Representations from models trained with finer-grained metadata encode topic information more effectively. When doing inference, only empty metadata is prepended, i.e. `<boc><eoc>`.

**Which parts of a URL matter for pretraining?** Unlike the other metadata types, URLs consist of different information. Taken `https://en.wikipedia.org/wiki/Metadata` as an example, we categorize `https://` as URL prefix, `en.wikipedia.org` as URL domain and everything after URL domain as URL suffix. Each component encodes different information, for example, URL domain usually corresponds to format and quality information, and URL suffix usually corresponds to topic information.

To investigate which parts of the URL contribute to the success of URL prepending, we sample 100 documents from Fineweb-Edu conditioned on their URLs and analyze attention patterns across layers. The absolute attention weights given to each part of the prepended components are plotted out in Figure 5, where it is observed that a significant portion of attention is directed toward the URL prefix — the part of the URL that carries no content information from the document. This observation highlights a common attention behavior: the model often focuses on consistent initial tokens, which can act as an "attention sink" without providing a meaningful signal for the task (Gu et al., 2025).

Since the majority of attention is given to the URL prefix part, does the attention pattern indicate that prepending the URL prefix should recover most of the URL-prepended performance? It turns out not to be the case. We conducted three ablation runs with different URL components prepended, as shown in Table 2. Adding only the prefix fails to surpass the standard baseline, even though its training loss trajectory closely follows that of the full-URL condition (Figure 13). We interpret this faster loss reduction as a *copying effect*: the suffix tends to act as a brief synopsis of the page and reveals part of the upcoming text, making next-token predic-

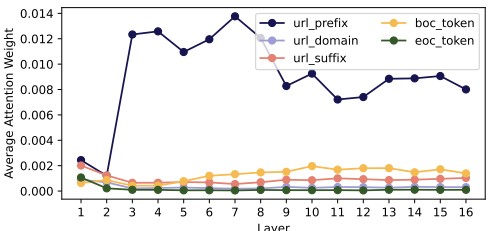

Figure 5: Averaged attention weights across all attention heads in different layers. A finer-grained per-layer per-attention head attention pattern is provided in Figure 6.

tion easier. URL domain and suffix, even though given relatively little attention to, are vital for the improvement in downstream tasks. Moreover, neither of them can catch the performance of full URL prepended run, suggesting that the domain and suffix encode complementary information. We offer more evidence and analysis in Appendix B.2.

> **Observation 2.** URL-prepended model shows a pronounced attention sink on URL prefix, but this does not contribute to the improved performance. In contrast, the URL domain and suffix provide complementary contributions.

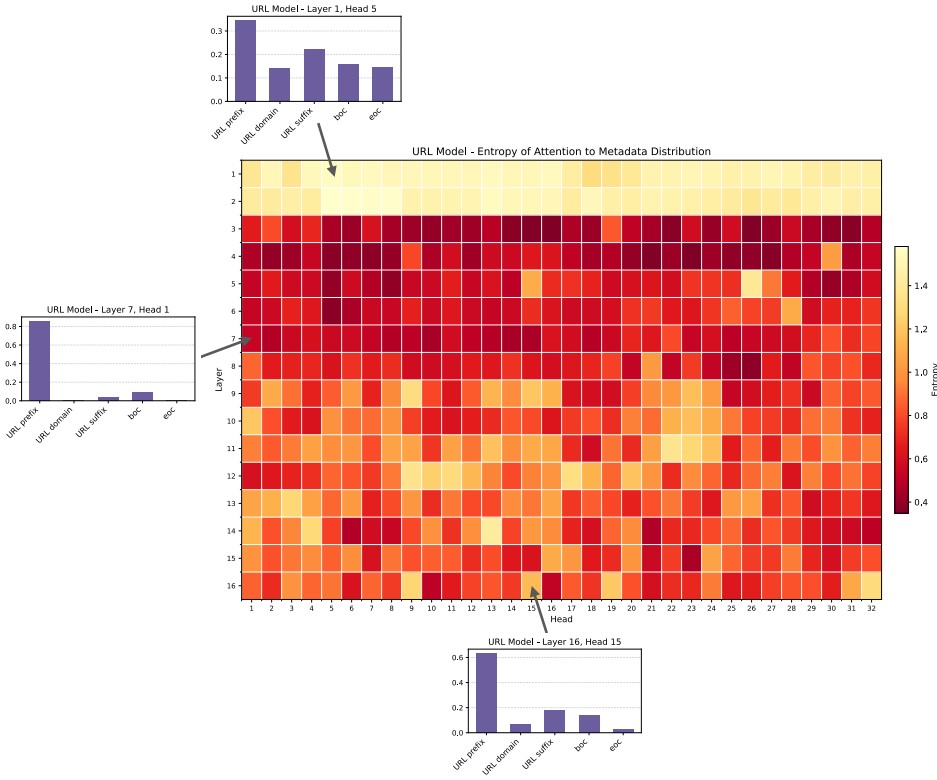

Figure 6: Entropy over the attention weight distribution to the different parts in the prepended context, plotted per layer and per attention head. Higher entropy denotes attention is more uniformly distributed, while low entropy indicates an attention sink.

Table 2: Downstream evaluation results for prepending various components of URL. Prepending URL domains provides a benefit, but it does not match the gains from prepending full URLs.

|  | Arc-C | Arc-E | CSQA | MMLU | PIQA | SIQA | HS | LBD | WG | Avg |
|---|---|---|---|---|---|---|---|---|---|---|
| URL | 34.8 | 71.7 | 46.9 | 32.5 | 72.3 | 41.5 | 42.5 | 32.4 | 55.7 | **47.7** |
| URL prefix | 34.0 | 68.9 | 43.0 | 32.1 | 72.0 | 41.7 | 42.0 | 32.8 | 53.1 | 46.6 |
| URL domain | 35.3 | 69.3 | 46.4 | 32.3 | 72.9 | 39.9 | 42.3 | 32.1 | 54.2 | 47.2 |
| URL suffix | 34.1 | 68.1 | 44.7 | 31.9 | 72.1 | 41.0 | 42.4 | 32.0 | 55.8 | 46.9 |

## 4.2 APPENDING: PREDICTING AUXILIARY INFORMATION MAY HELP

Beyond prepending, metadata can also be inserted at various positions during training. Khalifa et al. (2024) explored placing metadata at the beginning or end of a document, as well as repeating it multiple times within the text, for the purpose of data attribution. A natural question for improving pretraining performance is whether appending metadata could be beneficial.

Appending turns metadata prediction into an auxiliary task: after processing the entire sequence, the LM is asked to predict information such as the quality score or topic of the sequence. This setup may encourage the model to build an internal representation of the input that is sufficiently informative to recover the metadata at the end, potentially incentivizing it to compress salient aspects of the sequence into its hidden states. In this way, the auxiliary objective could serve as a form of soft regularization, providing another learning signal for the model weights.

We report the downstream performance for the five metadata types in the right panel of Figure 2, with the corresponding averages summarized in Table 1. Among these runs, appending fine-grained domain information helps the most. Additionally, coarse-grained quality score and URL appending also help improve downstream performance. In contrast, fine-grained quality score does not improve

over standard pretraining. The acceleration effect of helpful metadata is, on average, not as great as prepending; however, we are still able to train on 20% fewer tokens to achieve the same performance as standard pretraining.

> **Observation 3.** Auxiliary tasks such as coarse-grained quality score and fine-grained domain information prediction can accelerate training.

**Fine-grained vs. coarse-grained quality score.**  It is noteworthy that appending coarse-grained quality scores (single-digit integers in $3, 4, 5$) leads to better downstream performance than appending fine-grained quality scores (two-digit integers ranging from 25 to 50). In principle, a model trained with fine-grained scores could simply learn to predict the first digit and thereby achieve performance comparable to the coarse-grained case. However, it does not appear to do so. To verify this, we evaluated the fine-grained model's ability to predict the two-digit quality score by prompting it to continue generation after `<boc>`, and found that its predictions were highly accurate. We hypothesize that the model becomes overly focused on solving this auxiliary prediction task, which detracts from its ability to develop other skills that would improve downstream performance.

To evaluate this hypothesis, we run the probing experiments described in Section 3. Results for predicting document topic and quality appear in Figure 7. In the left panel where the task is unrelated to the included metadata type (quality), the QS-coarse appended model outperforms the QS-fine appended model, especially in the early layers. In contrast, in the right panel where the task is to predict quality, the QS-fine model performs slightly better. This pattern suggests the QS-fine model is over-specializing on the quality-prediction auxiliary task, with a corresponding trade-off in other capabilities.

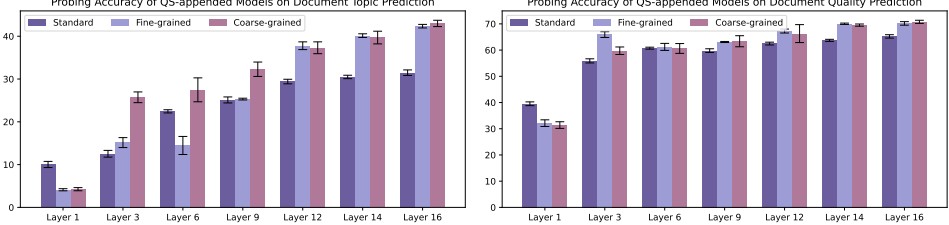

Figure 7: Probing results of two models that differ only in the granularity of the appended quality-score labels for predicting document topic and quality. The model with fine-grained quality scores performs better or on par with the coarse-grained model for document quality prediction, but performs worse on document topic prediction.

## 4.3 DOES ACCELERATION ALSO APPLY TO TRAINING LOSS?

How is the training acceleration reflected in the training curves? In the left panel of Figure 8, we show the training perplexity curves for all runs that outperform the standard pretraining. Overall, there appears to be no clear correlation between downstream performance and training loss. The only notable exception is URL prepending, which leads to a visibly faster decrease in loss, consistent with the findings of Fan et al. (2025). We attribute the faster loss reduction to a copying effect: URLs offer a shortcut for next-token prediction by enabling the model to simply copy tokens from the URL.

On top of this, we plotted out the gradient norm change throughout the whole training. Compared to the other runs that improve downstream performance, the standard pretraining run exhibits higher loss spikes. This suggests that including metadata may help stabilize LLM pretraining.

## 4.4 CAN LMS LEARN TO ACQUIRE META INFORMATION?

It is unsurprising that LMs can leverage additional metadata to infer latent clusters. A natural question is whether LMs can also infer such metadata on their own. To investigate this, we introduce five new meta tokens, `<s1>` through `<s5>`. These tokens do not exist in the original vocabulary and

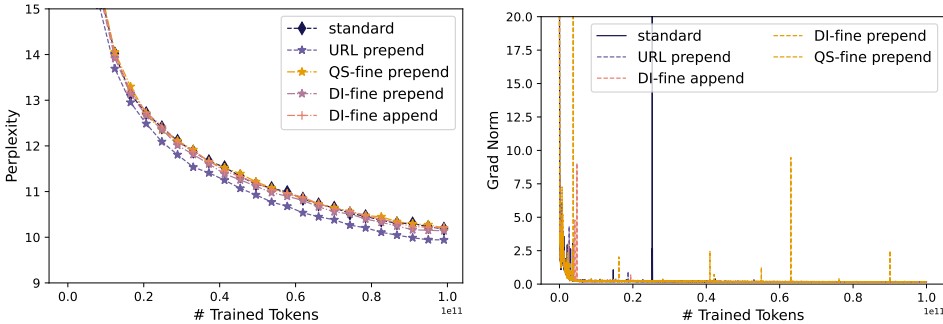

Figure 8: Training loss and gradient norm throughout the whole training.

are prepended to each sequence with probability 0.9, enclosed by `<boc>` and `<eoc>`. As all the prepending runs, we mask out the meta token loss for backpropagation.

With the five learnable meta tokens, we can observe an acceleration effect as well, as shown in the left panel of Figure 2. What information do the meta tokens encode? As the five meta tokens are always the same, we hypothesize that it is the attention to the five tokens that encodes useful information. To study this, we gathered synthetic documents of three different quality levels, and plotted out the average attention weights given to the five tokens for each quality level in Figure 9. Compared to the other low- and medium-quality documents, high-quality documents attend significantly less to `<s4>`.

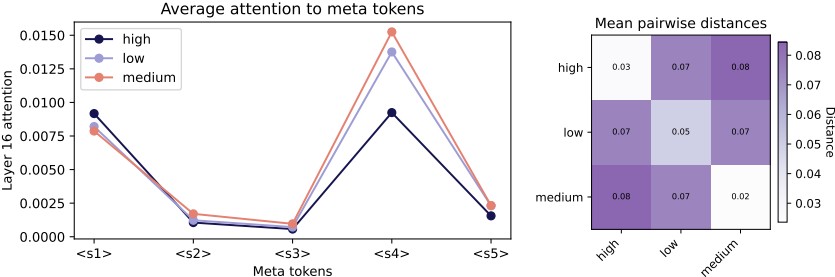

Figure 9: Attention pattern to the five prepended meta tokens (last layer). **Left**: average attention weights to each of the meta tokens; **Right**: inter- and intra-category attention pattern distance. Each category is a quality level and the distance is calculated via euclidean distance of stacked attention weights from the first 100 tokens of a document to meta tokens.

We further flatten the attention weights from the first 100 tokens of each document to meta tokens and compute the average Euclidean distances both within and across quality clusters. The results in Figure 9. show that inter-cluster distances are consistently larger than intra-cluster distances, indicating that documents of different quality levels exhibit distinct attention patterns. In contrast, similar experiments based on topic and format did not reveal such a clear separation in attention patterns across clusters, see Appendix B.1.

> **Observation 4.** LLMs can learn to encode quality-aware latent cluster information in learnable tokens that do not inherently carry any semantic meaning.

## 4.5 LATENT REPRESENTATION SHAPING

To better understand the differences in learned latent representations, we probe over three different tasks: writing style (approximated by authorship), document topic, and document quality. We present the results from an intermediate layer, as intermediate layers strike a balance between preserving signal and avoiding over-specialization (Skean et al., 2025).

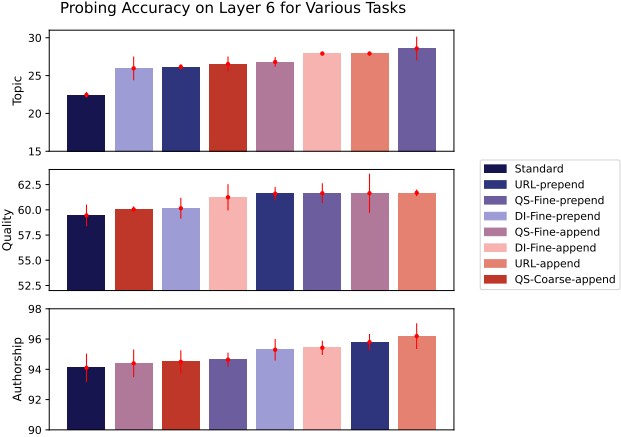

Figure 10: Probing accuracy across different model checkpoints on three tasks. In each configuration, we conducted 3 seed runs, and the standard deviation is reflected in the red bars.

For authorship, models enhanced with URL and DI-Fine metadata achieve the highest performance, suggesting that both types of metadata contribute to capturing writing style. Across all three tasks, the standard pretrained model shows the lowest probing accuracy, indicating a limited latent understanding of these higher-level concepts, as shown in Figure 10. For quality, the most effective metadata are URL and QS-Fine, reinforcing the idea that URL encodes information about document quality. In the case of document topics, the top-performing models are QS-Fine prepend, URL-append, and DI-Fine append, though no consistent pattern emerges across these results. We argue that the task is very challenging and the signal may be harder to observe.

> **Observation 5**. Including the URL as metadata (whether prepended or appended), enhances the model's latent grasp of writing style and overall document quality.

## 5 CONCLUSION

In this work, we show that LLM pretraining can be accelerated by conditioning on a wider range of metadata, extending beyond the commonly used URL. Our experiments indicate that fine-grained metadata, such as fine-grained quality scores and domain information, consistently yields greater improvements than coarse-grained alternatives when prepended. Appending metadata as auxiliary prediction tasks can accelerate training as well. We witness the biggest acceleration when fine-grained domain information is appended.

Overall, our findings highlight metadata as a versatile and underexplored lever for improving the efficiency and quality of LLM pretraining. An open question remains whether metadata can also enhance post-training. While we made initial attempts to explore how metadata shapes representations and gained some mechanistic insights into what aspects are improved, we still lack a clear understanding of why metadata is effective. We hope this work motivates the community to investigate these directions further.

**Acknowledgment.** This work was supported as part of the Swiss AI Initiative by a grant from the Swiss National Supercomputing Centre (CSCS) under project ID a06 on Alps. We acknowledge funding from SNSF Grant number 10005248.

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

## A   LLM USAGE STATEMENT

We used LLMs to polish writing as well as modify plotting scripts. Furthermore, we utilized LLMs for synthetic document generation in Section 4.4.

## B   MORE EXPERIMENTAL RESULTS

### B.1   ATTENTION PATTERN ON TOPIC AND FORMAT CLUSTERS

Additional experiments to Section 4.4. The attention pattern to meta tokens does not encode topic/format information, as we do not observe more similar attention patterns within one cluster in Figure 11, and Figure 12, respectively.

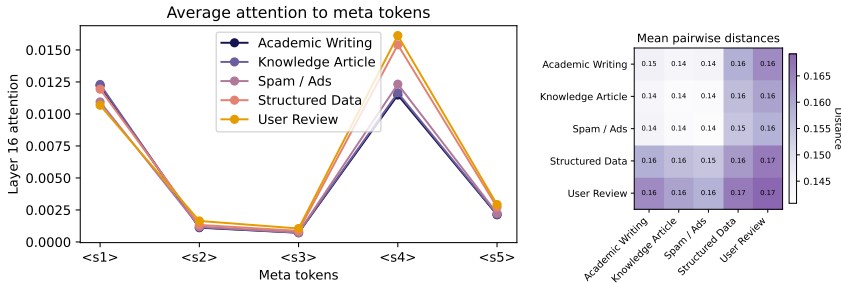

Figure 11: Attention pattern to the five prepended meta tokens (last layer). The documents are clustered by 5 different topics. Left: average attention weights to each of the meta tokens; right: inter- and intra-category attention pattern distance.

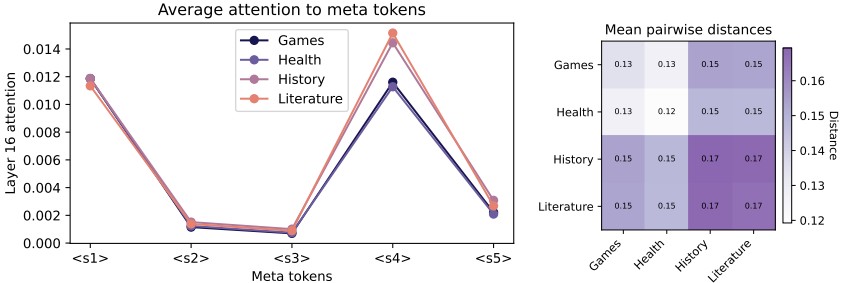

Figure 12: Attention pattern to the five prepended meta tokens (last layer). The documents are clustered by 4 different formats. Left: average attention weights to each of the meta tokens; right: inter- and intra-category attention pattern distance.

### B.2   FURTHER EXPERIMENTS ON UNDERSTANDING DIFFERENT URL PARTS

We extend the analysis from Figure 5 to all heads by aggregating attention into five categories: URL prefix, domain, suffix, boc, and eoc. We then normalize these values to obtain a probability distribution and compute the entropy. Low entropy values indicate a concentrated distribution that approaches a one-hot vector, signaling a significant attention sink for the URL prefix. These patterns are illustrated in Figure 6.

We also show the training loss curves for models prepending different portions of the URL in Figure 13. Notably, adding only the URL suffix yields a drop in training loss nearly as fast as when the full URL is prepended. We attribute this to a copying effect: the model can easily repeat tokens from the URL suffix, which often functions as a concise summary. However, relying solely on the suffix does not achieve the same strong downstream performance as using the full URL (as shown in Table 2), indicating that the domain and other upstream components of the URL provide additional useful signals.

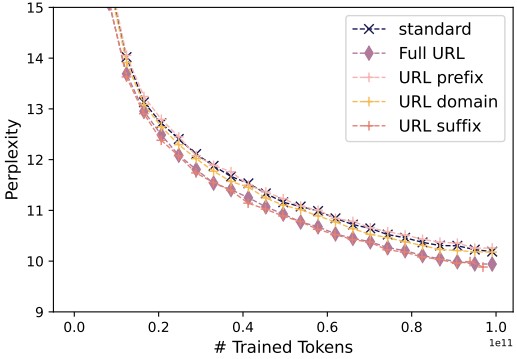

Figure 13: Training loss curves across different models with URL parts pre-conditioning. Both full URL and URL suffix enable a faster loss decrease.

### B.3 No Additive Effect from Prepending and Appending

Given that multiple prepended metadata types offer no cumulative benefit, we tested a hybrid approach: prepending and appending two different and helpful metadata types. We experimented with this by prepending URL and appending QS-coarse, which both can outperform standard pretraining individually. However, as shown in Figure 14, pairing both of these methods fails to surpass the performance of using a prepended URL by itself.

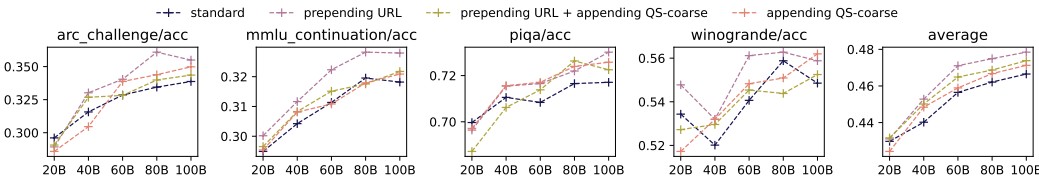

Figure 14: There is no additive effect for URL prepending and QS-coarse appending.

### B.4 Probing Accuracy for Other Models

To verify that our probing procedure is sound, we run a sanity check on latent representations from the Qwen2.5 model family (Qwen et al., 2025). The results are presented in Figure 15. For a model with comparable size but greater capability than ours, the probes achieve higher accuracy. Increasing the model scale further, from 1.5B to 7B, yields an additional boost in probing performance.

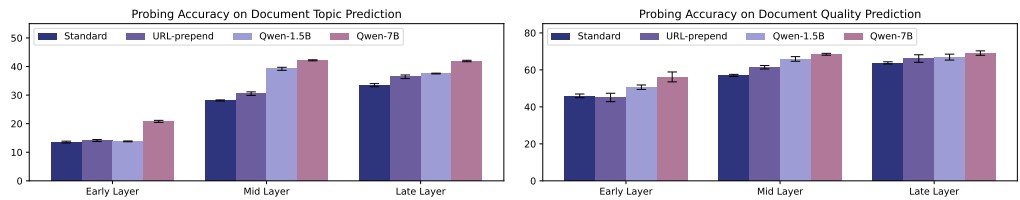

Figure 15: Comparison of probing accuracy for our standard, and URL-prepended model with Qwen 2.5 models for document topic and quality prediction models.

## C Prompt for DI-Fine generation

We used the Llama3.1-8B-Instruct model to annotate the documents for DI-fine models pretraining. The following prompt is used for the generation.

Based on the given sampled snippet from a document (it could be a
    webpage, book, codebase, paper, or anything else), write two concise
    keyphrases that together capture the document's domain:

TOPIC (less than 3 words) – the main subject matter.

FORMAT (less than 3 words) – the document's genre or source type.

Examples of valid outputs include:
quantum physics, research paper

healthy cooking, personal blog

video games, forum thread

*** Start of the snippet ***

\{snippet\}

*** End of the snippet ***

Now output only the two keyphrases in the exact form:

<TOPIC>, <FORMAT>

