# OpenReview forum: "Beyond URLs: Metadata Diversity and Position for Efficient LLM Pretraining"
_ICLR.cc/2026/Conference — ICLR 2026 Poster_

### Official Review · Reviewer_7BEe · 2025-10-25

**Soundness:** 4
**Presentation:** 3
**Contribution:** 3
**Rating:** 6
**Confidence:** 3

**Summary:**

- It is widely believed that prepending metadata helps LLMs learn latent cluster structure during pre-training; however, evidence beyond URL tags has been limited.
- The authors systematically investigate which metadata works best and find that fine granularity is key to accelerating LLM pre-training.
- Specifically, they compare several kinds of metadata (URL; coarse- vs. fine-grained quality scores and domain labels) and contrast prepending with appending.
- They show that fine-grained metadata yields larger gains than coarse-grained metadata, and that appending can also accelerate training.
- To probe the mechanism, they further analyze URL metadata: they observe an “attention sink” toward the URL prefix, yet the literal prefix content itself is not important.

Note: I used ChatGPT for minor language editing and phrasing assistance; all technical assessments are my own.

**Strengths:**

- They compare metadata across five configurations (varying granularity and including non-URL metadata) and evaluate both prepending and appending strategies.
- Their experimental analysis is multifaceted:
  - (i) They evaluate a broad suite of benchmarks;
  - (ii) They explore combinations of different metadata types;
  - (iii) They measure probing accuracy for latent cluster prediction;
  - (iv) They analyze attention scores and distances between attention patterns;
  - (v) They report perplexity and gradient norms.
- It is also interesting that QS-coarse model outperforms QS-fine model when the task is irrelevant to metadata.


Note: I used ChatGPT for minor language editing and phrasing assistance; all technical assessments are my own.

**Weaknesses:**

- The motivation for studying metadata granularity could be clarified further; otherwise, it risks seeming trivial to prefer fine-grained metadata whenever available. (See the question section.)
- The discussion in lines 347–353 may already address this concern; if so, please make this explicit—e.g., by emphasizing the key argument—and, if possible, provide any additional supporting rationale.

Note: I used ChatGPT for minor language editing and phrasing assistance; all technical assessments are my own.

**Questions:**

- Isn’t it trivial to prefer fine-grained metadata? Did you consider hypotheses under which (i) granularity does not matter, or (ii) coarse metadata might be preferable for some reason?
- It appears that [1] (which you cite in line 106) also compares metadata at different granularities—they state, “We compare the results by varying the depth of prepended metadata…” in Section 3. Do you see substantive similarities between your results and theirs?
- Can we expect that using both "prepending" and "appending" in the same training sequence would further boost pre-training?

[1] Higuchi, R., Kawata, R., Nishikawa, N., Oko, K., Yamaguchi, S., Kobayashi, S., ... & Suzuki, T. (2025). *When Does Metadata Conditioning (NOT) Work for Language Model Pre-Training? A Study with Context-Free Grammars.* arXiv:2504.17562.


Note: I used ChatGPT for minor language editing and phrasing assistance; all technical assessments are my own.

---

> ### Author Response · Authors · 2025-11-21
>
> Thanks for your insightful feedback! Regarding your raised questions, we address them as follows:
>
>
> **W1 & Q1 & W2**: 1) We would not necessarily think that it is trivial to prefer fine-grained metadata. If it is truly like that, then prepending two different metadata types should always be better than prepending only one, since arguably two different metadata types provide more fine-grained information. However, our empirical results do not support this. 2) coarse-grained metadata may be preferred in the appending experiments, which, as the reviewer mentioned, can be found in Line 339-353 Fine-grained vs. coarse-grained quality score. In the appending experiments, having a finer-grained auxiliary classification task can make the embeddings overly focus on the auxiliary task and detract it from developing other skills that would improve downstream performance.
>
> **Q2**: The work from Higuchi et al. studies a synthetic setup, where the metadata is the _deterministic_ grammar for generating the sequences. In our setup, our metadata is not the deterministic language generation rule, but certain _traits_ of the sequence itself. It is unclear if the results from Higuchi et al. generalize directly to our realistic setup.
>
> **Q3**: We did this experiment indeed, and the URL prepending + QS-coarse appending gives 47.4 in the downstream tasks, which is between URL prepending (47.7) and QS-coarse appending (47.1). Sadly, we did not see complementary benefits. We have added this result to Appendix B.4 in our _updated manuscript_.
>
> If you have any further questions or ideas, please let us know. We’d be glad to share additional supporting information.

---

> > ### Comment · Reviewer_7BEe · 2025-11-26
> >
> > Thank you for addressing my questions on the motivation relative to prior work and on the prepend/append setting. The clarifications resolve my concerns. I will keep my positive score unchanged.

---

### Official Review · Reviewer_9nY6 · 2025-10-30

**Soundness:** 3
**Presentation:** 3
**Contribution:** 2
**Rating:** 6
**Confidence:** 3

**Summary:**

This paper presents a systematic empirical study of how different metadata types (URL, quality, domain) influence LLM pretraining.


Through controlled pretraining runs and a suite of probing and attention analyses, the paper finds that fine-grained metadata consistently accelerates learning and shapes latent representations, while coarse metadata contributes little.

Notably, URL metadata enhances stylistic and quality-related features but also introduces attention-sink behavior.

The work further explores “learnable meta tokens” as a self-organizing latent conditioning mechanism.

The study is systematic, clearly written, and practically relevant to LLM data curation pipelines.

**Strengths:**

1. The paper provides a broad and well-controlled comparison across five metadata types and multiple usage paradigms (prepending, appending, prediction).

2. Demonstrates that fine-grained metadata yields measurable pretraining efficiency gains, informing real-world LLM data curation.

3.  Includes diverse diagnostic views: loss curves, gradient stability, attention visualization, and probing of latent representations.

4. Writing and figures are clear; each section presents concrete “observations” that summarize key findings.

**Weaknesses:**

1. Limited robustness under real web-scale noise.

    - The paper’s findings rely on moderately curated corpora (e.g., FineWeb-Edu), where metadata fields are clean and semantically aligned with the text.
    - On truly raw web data, where a large fraction of pages contain boilerplate, encoding errors, or misaligned metadata, the assumed correlation between metadata and content weakens.

2. The learnable meta-token experiment is under-specified and likely reflects optimization or statistical effects rather than true semantic abstraction.
    - Since the tokens are inserted randomly and unsupervised, any observed clustering by “quality” could simply result from correlations between quality scores and superficial statistics such as document length or domain frequency, rather than genuine latent metadata inference.

3. Potential artifact in attention-sink analysis.
    - The paper attributes performance differences to an “attention sink” on URL prefixes, but the phenomenon is likely positional rather than semantic.
Because all metadata are prepended, prefix tokens naturally dominate early-layer attention regardless of content.
An append or randomized-position control would likely remove this effect. Without such controls, the claim remains unsubstantiated.

**Questions:**

1. Lack of quantitative definition of metadata informativeness
    - The paper repeatedly claims that fine-grained metadata is more beneficial than coarse-grained metadata, yet this distinction is treated qualitatively. There is no quantitative measure of metadata informativeness (e.g., entropy, number of distinct buckets, mutual information with the text, or token-level perplexity gain).
Without defining an “information budget,” it is difficult to generalize the conclusion or to predict when a given metadata type will be helpful. A systematic analysis relating metadata information content to training acceleration would make the claims much stronger.

2. Over-smoothed attention analysis
    - The attention analysis reports only average attention weights aggregated across all heads and layers. Such averaging may mask the presence of a few specialized heads that truly focus on useful metadata components (e.g., URL domain), while most others attend to superficial tokens like the prefix.
A head-wise or layer-wise breakdown would clarify whether meaningful metadata utilization emerges in specific submodules rather than being uniformly weak across the network. As it stands, the conclusion that “prefix attention is a sink and unhelpful” may be an artifact of excessive averaging.

---

> ### Author Response · Authors · 2025-11-21
>
> Thank you for the helpful feedback. We address each point below, using Wi for weakness i and Qi for question i
>
> **W1**: We offer a different perspective: improving performance on a highly curated dataset like FineWeb-Edu is generally more challenging than on raw data due to diminishing returns. Since the text quality is already high, the 'signal' is strong and the model relies less on external context to resolve ambiguities. The fact that metadata provides gains even in this high-signal regime suggests that it captures orthogonal information not present in the text, a benefit we expect to hold (or potentially increase) when the textual signal is noisier.
>
> **W2**: Thanks for the comment. However, we see those superficial statistics as an indication of document quality. For example, low-quality documents have a lot of verbose, rambling text with intentional repetition; medium-quality documents feature more concise and generic information; high-quality documents are usually ​​long texts with structured, comprehensive content.
>
>
> **W3**: We did not claim that performance differences stem from an “attention sink” on URL prefixes. As shown in Table 2, adding only a prefix does not reproduce the gains achieved by prepending the full URL. Instead, we are hypothesizing that the little attention to the URL domain and suffix parts contributes to the superior performance.
>
> **Q1**: We have provided the number of distinct buckets for types of metadata (see lines 148 to 161).  We restate them in the following table for your convenience. For the DI-fine case, since we use Llama model for open-end annotation, the label space is effectively unbounded. To answer your question, it is not the number of distinct buckets that contributes to the effectiveness of metadata inclusion.
>
> Domain-related signals capture something distinct from document quality, and the two operate in different ways. As shown in [1], domain information guides text generation better than the other metadata types, while [2] demonstrates that adding quality markers enables the model to recognize high-quality data automatically. For these reasons, a single information-richness threshold is unlikely to be effective. It may be necessary to define separate thresholds for different metadata types.
>
> | Metadata Types        | # Distinct Buckets           |
> |-------------|------------------------|
> | URL      |   unbounded    |
> | QS-coarse   | 3  |
> | QS-fine     | 35 |
> | DI-coarse | 576 |
> | DI-fine | unbounded |
>
> [1] Fan et al. URLs Help, Topics Guide: Understanding Metadata Utility in LLM Training
>
> [2] Allen-Zhu and Li. Physics of Language Models: Part 3.3, Knowledge Capacity Scaling Laws
>
>
> **Q2**: We have newly added a more detailed analysis in the Appendix regarding the per-layer head-wise attention patterns in the Appendix, see Figure 13.  To keep the visualization clear, we compute the entropy of the normalized attention distribution (over URL components and \<boc>\<eoc>). High entropy indicates evenly spread attention, while lower values signal attention sink over the URL prefix.
>
> Our observations are as follows:
>
> - In the first two layers, the meaningful components of the URL, such as the domain and suffix, receive a significant amount of attention.
>
> - In layers three through eight, all heads exhibit a pronounced attention sink toward the URL prefixes.
>
> - In the final layers (nine through sixteen), some heads consistently allocate noticeable attention back to the meaningful URL segments.
>
> If you have any further questions or ideas, please let us know. We’d be glad to share additional supporting information.

---

### Official Review · Reviewer_1P2e · 2025-11-01

**Soundness:** 2
**Presentation:** 2
**Contribution:** 2
**Rating:** 4
**Confidence:** 3

**Summary:**

This paper is a thorough study on the effects of conditioning *or* predicting metadata for pre-training documents. By means a number of controlled pre-training runs, the authors study the effects of different types of metadata and contrast conditioning and predicting. The main finding is that more granular data leads to more benefits. The paper proceeds to consider attention sinks and probing of internal states to understand the representational benefits of metadata learning.

**Strengths:**

* The paper presents a comprehensive analysis around a relatively understudied type of pre-training technique, metadata conditioning.
* The paper introduces a new technique, metadata prediction, and finds that it also provides benefits.
* The pre-training scale of the paper (1.5B runs with up to 100B tokens) is quite extensive.
* The paper takes a first step to build a more mechanistic understanding of the benefits of metadata conditioning by probing hidden representations.

**Weaknesses:**

* While the paper adds more evidence to Observation 1 (need for fine-grained granularity), the hypothesis was already formed and supported by some ablations by Gao et al., Metadata Conditioning Accelerates Language Model Pre-training.
* The paper should attempt to quantify the variance in the pre-training results and evaluations. I am little skeptical that the takeaways are all statistically significant. Specifically, the results in Figure 3 are worrying, since information-theoretically, prepending two types of information should yield similar benefits. (Unless it leads to substantially fewer "predicted" tokens during training)
* The insights in Observations 2, 3, and 4 are interesting, but rather specific and anecdotal, such that the wider relevance and applicability of these insights is not clear to me. The paper also makes limited progress in my opinion towards a foundational explanation of why metadata conditioning leads to improved pre-training results.

**Questions:**

Do you think the attention sink effect of metadata conditioning is an important benefit of the technique?

Can metadata conditioning and metadata prediction be combined to yield complementary benefits (for two different types of metadata information)?

What would be the recommendation of the paper with regards to best practices for metadata conditioned pre-training?

---

> ### Author Response · Authors · 2025-11-21
>
> Thank you for the helpful feedback. We address each point below, using Wi for weakness i and Qi for question i
>
> **W1**: Upon reviewing the MeCo paper, our interpretation differs regarding the hypothesis on metadata granularity. We noted that the authors wrote, 'absolute domain names (e.g., en.wikipedia.org) provide the appropriate granularity as metadata,' suggesting that coarser-grained URL domains are sufficient.
>
> To ensure we are not overlooking a crucial detail, could you kindly point us to the specific part of the paper that discusses the finer-granularity hypothesis? This would help us address your concern precisely.
>
> **W2**: We agree with you that it is important to account for the variance in the results. In fact, for _all the runs that lead to a better averaged performance compared to standard training_, the scores we reported are averaged across 3 seeds. We wanted to make sure the enhanced performance is consistent.
>
> We also find Figure 3 a bit surprising –  we also expected the LM to perform at least as good as URL prepending, as it can simply learn to ignore the QS-fine part; however existence of such a model does not guarantee learnability. We suspect the interaction between two distinct metadata signals can introduce noise or conflicting signals that make it harder for the learning algorithm to converge to the optimal solution compared to the single-metadata case. The model does not automatically act as a selector; it attempts to integrate all available context, which can lead to interference.
>
> **W3**: We recognize that we have not proposed a fundamental explanation for why metadata conditioning works, as is the case for all related studies. Although we examined the representation space and attention behaviors in depth, we did not uncover convincing correlational patterns, let alone causal ones. As a result, rather than offering a definitive conclusion, our work contributes evidence that sheds light on parts of the underlying mechanisms. We respectfully argue that Observations 2, 3, and 4 are valuable because they represent a comprehensive investigation of the problem "from every angle," rather than just anecdotal findings
>
> We also underscore that our study is the _first_ in this line of research to present a thorough and original investigation of (1) different metadata placements, (2) the use of empty metadata tokens, and (3) attention patterns and probing analyses. We believe these contributions represent a meaningful advance for research in this area.
>
> **Q1**: We do not think so. The attention sink phenomenon also happens with other types of metadata, for example, with quality score prepending, we always prepend “Quality Score: \<quality_score>” to each document, and we noticed attention sink to “Quality”, “Score” and “:”. However, this does not give the same performance boost as URL prepending.
>
> **Q2**: We did this experiment indeed, and the URL prepending + QS-coarse appending gives 47.4 in the downstream tasks, which is between URL prepending (47.7) and QS-coarse appending (47.1). Sadly, we did not see complementary benefits. We have added this result to Appendix B.4 in our updated manuscript.
>
> **Q3**: Our best recommendation is still URL prepending for pretraining data efficiency. If one wants better embeddings (for downstream classification tasks, for example), we would recommend metadata appending.
>
> We hope this clarifies our findings. If you feel that our response sufficiently addresses your concerns, we kindly request that you consider raising your score. If any outstanding issues remain, please let us know, and we'll be happy to provide further clarification.

---

### Official Review · Reviewer_QQyK · 2025-11-03

**Soundness:** 3
**Presentation:** 4
**Contribution:** 3
**Rating:** 8
**Confidence:** 4

**Summary:**

This paper extends the prior work on using URLs to accelerate pre-training to add additional metadata information. The experiments are with a 1.5B LLaMA model on FineWeb-Edu (which I think the authors should amend the abstract to mention upfront). They also study whether it's better to append or prepend the metadata. Both provide acceleration but it seems like appending might be better for building a more rich representation space. They abstract away concrete metadata entirely by providing learnable meta-tokens that have no semantic meaning but encode quality-related structures in the attention patterns.

**Strengths:**

1. The space of data augmentation via metadata is underexplored and quite promising for accelerating pre-training with negligible extra computational cost.
2. It is interesting to see more interpretability analyses on what metadata does in the model. The connection to attention sink is especially interesting.
3. Experiments and ablations are run well and conducted thoroughly. The paper is written well and easy to understand.

**Weaknesses:**

There is a lot of speculation around what the metadata does and it does not have clear grounding in empirical results. First, the optimization speedup is hard to understand and isn't described quantitatively. Figure 7 provides little insight into it -- I am especially unsure what to take away from the gradient norm, given the lack of other information (gradient moment estimates, update norms, etc).

There are also not enough hyperparameter ablations (I know they are expensive) to draw clean conclusions about the benefit of metadata wrt optimization. For example, Section 4.2 speculates about a soft regularization but it is hard to understand what that truly corresponds to -- for example, is it an optimization or generalization benefit? Both? The text is too vague on this point and I think there are not enough experiments to make claims of this type.

The interpretability studies are interesting but done too coarsely (eg averaged across all heads and layers). The authors may want to expand the appendix to detail more information for this to be a useful interpretability study to others.

**Questions:**

See above

---

> ### Author Response · Authors · 2025-11-21
>
> Thanks for your encouraging and insightful feedback.
>
> Regarding the raised weaknesses, we address them as follows:
>
> **W1**: We found that training loss does not necessarily correlate with downstream performance. We newly added Figure 12 in the _updated manuscript_, where we plotted out the perplexity when different parts of the URL are prepended for training. URL-suffix prepending gives an almost overlapping training loss as full URL prepending; however, its downstream performance cannot catch up with full URL prepending.  We believe that the lower training loss is due to the copying effect, i.e. URL suffix section often contains a one-sentence summary, and the LM can often copy words verbatim from it for next token prediction (see Appendix B.3).
> We present the gradient norm figure to show that it seems that metadata inclusion can help with suppressing gradient norm spikes. We suspect that the LM can learn to focus on the “good” data and downweight the outlier batches.
>
> **W2**: We apologize for the ambiguous wording. We wanted to say that the auxiliary prediction task provides another learning signal for the model weights, similarly to a regularization term. Even though the training loss remains at a similar value, the arrived optimum is better at predicting this auxiliary signal, suggesting that the benefit comes from the generalization effect of the metadata.
>
> **W3**: Thank you for the suggestion. We’ve added a detailed, layer-by-layer, head-wise analysis of attention patterns in Figure 13 of the _updated manuscript_. To keep the visualization clear, we compute the entropy over the normalized attention distribution (over URL components and \<boc>\<eoc>). High entropy indicates evenly spread attention, while lower values signal attention sink on the URL prefix.
>
> Our observations are as follows:
>
> - In the first two layers, the meaningful components of the URL, such as the domain and suffix, receive a significant amount of attention.
>
> - In layers three through eight, all heads exhibit a pronounced attention sink toward the URL prefixes.
>
> - In the final layers (nine through sixteen), some heads consistently allocate noticeable attention back to the meaningful URL segments.
>
> If you have any further questions or ideas, please let us know. We’d be glad to share additional supporting information.

---

### Author Response · Authors · 2025-11-21
**Global response**

Dear Reviewers,

Thank you for your time and thoughtful feedback. In response to your comments, we have revised the manuscript to include additional visual evidence and clearer explanations. All changes are highlighted in purple. We kindly invite you to review the updated version, and we hope it addresses your concerns.

Best,

Submission 20539 authors

---

### Meta-Review · Area_Chair_9aFc · 2026-01-06

**Summary:**

The paper studies the possibility of metadata conditioning speeding up LM pre-training. It identifies additional signals beyond the URL that can be useful, and offers some hypotheses (with accompanying evidence, of varying degrees) of why these signals can help training.

Reviewers generally appreciated the systematic empirical approach, while noting a few limitations:
- **Scope & grounding of observations**. Multiple reviewers (including those with positive ratings) noted that there are gaps between the main observations and the empirical results; and that the scope of the observations may be narrow. The central question of _why_ metadata conditioning can help pre-training was seen as remaining largely open, although the results of the paper offer _hints_ towards possible explanations.
- **Confounders in empirical results**. One reviewer noted that there are possible confounding effects in the attention sink and learnable meta-token results, which could qualify any conclusions about the role of metadata.
- **Relation to prior work**. Multiple reviewers requested clarification of the relation to prior works on metadata conditioning: Higuchi et al., _When Does Metadata Conditioning (NOT) Work for Language Model Pre-Training? A Study with Context-Free Grammars_; Gao et al., _Metadata Conditioning Accelerates Language Model Pre-training_.
- **Restriction to curated data**. Multiple reviewers noted that the results are restricted to the FineWeb-Edu pre-training corpora, and the transfer to messier corpora is unclear.
- **Statistical significance**. One reviewer noted that the gains with metadata may be within the expected noise range.
- **Granularity of interpretability studies**. One reviewer critiqued the interpretability study as operating on too coarse a granularity as to draw robust conclusions.
- **Motivation for metadata granularity**.  One reviewer requested clarification of the motivation for studying the role of metadata granularity, and why this was not trivial.

**Reviewer Concerns:**

- **Scope & grounding of observations**. The authors emphasized that despite their work not fully resolving the precise reasons behind metadata being valuable, they provide a comprehensive study of this issue from multiple angles. The authors also provided further explanation for the observations in Figure 7.
  - *Partially addressed*. The discussion of Figure 7 makes sense. There seems to be agreement that there are some limits to the empirical grounding of stated hypotheses remains. The question is thus whether the provided results are sufficiently interesting by themselves to help future work. From our reading, we are in favor of this view, but only marginally.
- **Confounders in empirical results**. The authors argued that their claim around attention sinks is not that they improve performance; rather, it is that _despite_ URL prefixes having a high attention mass, they alone do not contribute to good performance (per Table 2). The authors also argued that for the learnable meta-token results, superficial document statistics may themselves be an indication of quality.
  - *Partially addressed*. The authors' argument on attention sinks appears in keeping with Section 4.1. However, the reviewers' suggestion for randomization of the position should be considered. Regarding the discussion on document quality and learnable meta-tokens, the core claim in Observation 4.4 appears well defended, with the debate being around the interpretation of these meta-tokens. We see the reviewer's conjecture of the correlation being to superficial features as worthy of further analysis. This could motivate a slight change in the wording of Section 4.4.
- **Relation to prior work**. The authors noted that the key distinction to the referenced work of Higuchi et al. is the use of deterministic grammar rules, versus naturally occurring metadata. They argued that it is unclear that results for the former should necessarily translate to the latter. In relation to Gao et al., the authors argued that the work considers coarse-grained granularity, and does not explicitly consider the distinction to fine-grained granularity.
  - *Mostly addressed*. The authors' argument appears reasonable. It would be prudent to add a more explicit discussion of this prior work in an updated version.
- **Restriction to curated data**. The authors argued the gains on FineWeb-Edu should be seen positively, as here the high text quality implies a lower headroom for gains through further conditioning. The authors argued that one should expect larger gains on noisier data.
  - *Partially addressed*. The authors' argument does make sense. However, it would be most compelling if there were at least some empirical results validating their hypothesis. As the reviewer noted, with raw web data, it is unclear what overall correlation is expected between the metadata and page content. As it stands, it remains open to what extent the results presented will transfer to other settings.
- **Statistical significance**. The authors reported that all runs involving superior performance to standard training were from 3 independent trials. The authors also argued how Figure 3 could be reasonable, given possible conflicts between the two individually beneficial signals.
  - *Partially addressed*. A clear discussion of the experimental protocol, and standard deviation across trials, appears missing. The hypothesis about conflicting signals in Figure 3 is plausible, but seems a natural avenue for further analysis (as done in the rest of the paper).
- **Granularity of interpretability studies**. The authors added per-layer and per-head analyses of the attention patterns in Fig 13. These evince that attention sinks are pronounced in middle layers, while in early layers the other URL components receive more attention mass.
  - *Mostly addressed*.
- **Motivation for metadata granularity**. The authors noted that their empirical results refute the intuition that fine-grained metadata should always help, noting that this may be due to the model becoming overly specialized to the auxiliary task.
  - *Mostly addressed*. The authors' argument appears reasonable. It would be prudent to add a more explicit discussion of this in the Introduction of an updated version.

**Reviewer Scores:**

- **9nY6**: the reviewers' comments were reasonably responded to, but there can still be outstanding questions (e.g., precisely what the performance is on datasets other than FineWeb-Edu). Thus, we think it likely the score would remain at 6.
- **QQyK**: as the review was overall positive without major qualifiers, we believe it likely the score would remain at 8.
- **7BEe**: the reviewer mentioned being satisfied with the response, and keeping their positive score of 6 unchanged.
- **1P2e**: per above, we see the reviewer's comments as being only partially addressed. We think it likely to remain at 4, though it is plausible that the score could increase to 6,.

---

### Decision · Program_Chairs · 2026-01-26

Accept (Poster)